# Gibbs Energy and Gene Expression Combined as a New Technique for Selecting Drug Targets for Inhibiting Specific Protein–Protein Interactions

**DOI:** 10.3390/ijms241914648

**Published:** 2023-09-27

**Authors:** Edward A. Rietman, Hava T. Siegelmann, Giannoula Lakka Klement, Jack A. Tuszynski

**Affiliations:** 1Manning College of Information and Computer Science, University of Massachusetts, Amherst, MA 01003, USA; erietman@gmail.com (E.A.R.); hava.siegelmann@gmail.com (H.T.S.); 2Applied Physics, 477 Madison Ave., 6th Floor, New York, NY 10022, USA; 3CSTS Healthcare 403 Melita St., Toronto, ON M6G 3X2, Canada; giannoula@aaiomics.com; 4Department of Mechanical and Aerospace Engineering, Politecnico di Torino, I-10129 Turin, Italy; 5Department of Data Science and Engineering, The Silesian University of Technology, 44-100 Gliwice, Poland; 6Department of Physics, University of Alberta, Edmonton, AB T6G 2E9, Canada

**Keywords:** protein–protein interaction, PPI, KEGG, TCGA, chronic lymphocytic cancer, glioma

## Abstract

One of the most important aspects of successful cancer therapy is the identification of a target protein for inhibition interaction. Conventionally, this consists of screening a panel of genes to assess which is mutated and then developing a small molecule to inhibit the interaction of two proteins or to simply inhibit a specific protein from all interactions. In previous work, we have proposed computational methods that analyze protein–protein networks using both topological approaches and thermodynamic quantification provided by Gibbs free energy. In order to make these approaches both easier to implement and free of arbitrary topological filtration criteria, in the present paper, we propose a modification of the topological–thermodynamic analysis, which focuses on the selection of the most thermodynamically stable proteins and their subnetwork interaction partners with the highest expression levels. We illustrate the implementation of the new approach with two specific cases, glioblastoma (glioma brain tumors) and chronic lymphatic leukoma (CLL), based on the publicly available patient-derived datasets. We also discuss how this can be used in clinical practice in connection with the availability of approved and investigational drugs.

## 1. Introduction

Cancer is one of the most challenging diseases worldwide because its complexity is intricately bound to the physiological deregulation of ourselves. Although a significant improvement in diagnosis and treatment has occurred in the past few years, progress has been slow, and the rate of death should reach a staggering 13.2 million deaths worldwide by 2030 [1], a picture that is expected to become worse in the future as a result of the general trends of population aging [2].

The generally accepted strategy for cancer treatment is to remove the solid tumor by surgery, if possible, and then to target malignant cells with radiation therapy, focusing on the tumor site. Additionally, chemotherapeutic approaches with systemic and targeted modes of delivery are commonly used with the highest possible level of specificity and selectivity of the pharmacological agent or a combination of drugs. A drug may be designed to bind to general targets, such as DNA replication mechanisms or nucleotide synthesis machinery of the cell since malignant cells typically divide faster than normal cells and abolish pro-apoptotic signaling. Chemotherapy agents also commonly inhibit targets that belong to metabolic or protein–protein interaction (PPI) networks. The proper identification of molecular mechanisms that drive tumorigenesis and cancer progression is crucially important for providing efficacious therapeutics since tumorigenesis can be viewed as the dysregulation of protein–protein interaction networks that control the cellular function, especially growth and division. However, what complicates this approach is the presence of redundancies and complex feedback loops that provide robustness in this vast network of interacting proteins. In the language of network theory, this complex architecture confers the integrity of the entire system in the case of vertex (protein) or edge (interaction) failure [3]. As a consequence of this built-in stability, malignant cells have a tendency to resist or evade the effects of a drug by switching to an alternative metabolic or signaling pathway whenever possible. Therefore, for most PPI networks, the removal of a randomly chosen vertex by a drug (for example, due to ubiquitination caused by a PROTAC entity) may not produce the desired result in terms of the performance of the entire PPI network. Conversely, targeting deliberately chosen vertices functioning as hubs in a network may cause its break up into fragments, ultimately leading to cell lethality. Thus, targeting vertices with high connectivity and betweenness centrality is expected to significantly improve the odds of a successful therapeutic outcome of drug or drug combination effects on malignant cells [4]. Recently, numerous potential molecular targets for cancer therapy (oncotargets) have been identified, leading to the design and development of pharmacological agents with inhibitory properties with respect to these targets. However, most of the currently available drugs are very expensive and provide modest improvements in objective survival, which is coupled with significant adverse side effects. Within the above context, new strategies are needed to carefully and insightfully investigate the pros and cons of specific molecular targets that may or may not result in beneficial outcomes for the patient from the administration of drugs. Consequently, the focus should be placed on those targets that (i) are involved in physiological dysregulation, leading to cancer initiation, (ii) act as hubs for signaling pathways, leading to proliferation and survival stimuli through positive feedback loops in the PPI networks [5], and (iii) increase the risk of metastasis development.

Fortunately, at present, the knowledge of the interactions between cellular proteins is now sufficiently well developed at the molecular level so that modeling complex molecular processes can be performed at a high level of confidence. Protein interactomes have been well characterized by the use of refined experimental techniques, such as two-hybrid yeast, affinity pull-down mass spectrometry, biochemical techniques, and next-generation sequencing. The massive amount of resultant data have been made available through online databases. Furthermore, recent progress in data mining, high throughput data generation relative to gene, protein, and metabolic networks, as well as computer simulations of protein–protein interactions [4,6], offers a new and very attractive opportunity to identify those proteins that would have marginal significance in normal cells but would become signaling hubs in cancer cells. This would be due to the high level of connectivity of the target proteins with other proteins, such that a significant modification of expression levels would strongly affect cellular survival.

Complex networks are very common in the life sciences, at the level of system descriptions ranging from genes to proteins to cells and also to organisms and even societies. Mathematically, a network is typically described by a directed or undirected graph G = (V, E) with vertex and edge sets V and E, respectively. An edge is allocated to the graph where there is a known interaction between two elements, each of which corresponds to a distinct vertex. The corresponding interaction represents either the direct binding of the two interacting elements, the functional activation of one element by another, or the initiation of enzymatic catalysis. Real-world networks demonstrate a modular structure whereby subsets of vertices are organized in clusters when they are tightly connected internally and loosely connected to other vertices outside the given cluster [7]. This organization leads to the emergence of symmetric subgraphs, such as trees and complete cliques [5], which often facilitates the classification of the network’s vertices into its backbone and appendages.

Within the scope of applications of PPI modeling in the context of cancer chemotherapy, early model development provided a link between the protein level of description and cancer epidemiology [4,6]. Specifically, it has been shown that the probability of 5-year patient survival, using the database of Surveillance Epidemiology and End Results—SEER for the main types of cancer, is negatively correlated with network entropy [4,6]. As a measure of the network’s complexity, Shannon entropy derived for each topology of the corresponding PPI network was used together with the betweenness centrality computed for each vertex (protein) as an indication of its importance in the network. The following formula was used to calculate the network entropy for each type of cancer considered
(1)$$H=-\sum_{V=1}^{n-1}p\left(V\right)logp(V)$$
(2)$$C_{B}\left(V\right)=\sum_{s,t\in V}\sigma \left(s,t|V\right)/\sigma (s,t)$$
where σ(s, t) is the number of shortest paths between two nodes (s, t) and σ(s, t|V) is the number of those paths passing through nodes other than the (s, t) pair [4].

Typically, cancer patients are treated with the maximum tolerated dose (MTD) of some chemotherapeutic agent that is designed to target so-called oncogenes and/or microtubule dynamics. Often, the chemical agent is designed to target genes that are overexpressed. Usually, there is little regard for the protein coded for by the gene and its biochemical and/or cellular function. In our previous work, we have been exploring the use of Gibbs energy and protein–protein interaction (PPI) network topology for selecting protein targets for inhibition (i.e., inhibiting the activity of a specific protein target). Specifically, PPI networks involved in the signaling pathways that are over-expressed in a given type of cancer were analyzed. Briefly, the approach we have used, as described in the literature [8,9], is to take the complete mRNA expression of biopsied tissue, then overlay it on the complete human protein–protein interaction network [10], followed by calculation of the Gibbs energy contribution from each protein. We then performed a so-called topological filtration on the energy network to generate a subnetwork of important contributors to the Gibbs energy. After that, we compute the influence of each protein on the overall Betti number (a topological measure of network complexity). The selected protein target, to inhibit its reactivity with neighboring proteins, is the one that reduces the Betti number the most and, thus, the complexity of the energetic subnetwork. This approach became a new method for personalized medicine [8,9]. Extending that now to PPI inhibitions is the main focus of this paper. As pointed out by Keskin et al. [11], an important focus in studying and understanding protein–protein interactions is to predict how proteins interact with each other. In order to do this, their 3D structure must be known in great detail. Which now is possible thanks to deep neural networks [12,13,14]. Of course, the Lipinski “rule of five” will still be applicable for the discovery of molecules to block specific PPIs [15].

Finding good PPI targets to interrupt is a challenge. For example, [16,17] suggests that a good cancer target would be the interaction MDM2-p53. However, a recent review [18] points out that despite the number of clinical trials, none have been approved for clinical use, and [19] describes the challenges of in silico workflows using, as an example, MDM2-p53. More recent reviews on the computational discovery of which PPI to interrupt are given by [20,21,22]. One would ideally need to computationally explore a superposition of small protein–molecule interactions. As described by [23], this could be most easily performed using a quantum reservoir computer [24]. However, we suspect that an approach similar to the one presented in [25] should prove to be interesting.

In the following sections, we describe a new approach to select specific PPIs based on Gibbs free energy and subnetwork interaction partners. If our new technique is used with the human foldome (i.e., the known folded 3D structure of the set of human proteins to atomic precision), it is now possible to contemplate truly personalized medicine [12,13,14]. In the following sections, we present some results for glioma and CCL patients. Then we present the discussion of the results, followed by the materials and method. Lastly, we present our conclusions.

## 2. Results

Now, let us look at a real-world example of 514 brain cancer mRNA samples. The data are from glioma patients from the TCGA database [26]. After merging the relevant KEGG networks (described in the Materials and Methods section), the final resulting network has 144 nodes and 517 edges. By using the mRNA data for computing each protein’s contribution to Gibbs energy, followed by searching the neighbors for maximum expression, and doing this for each patient, we obtain the results shown in Figure 1.

Using the same methodology, we constructed Figure 2, which shows the most significant protein–protein interaction targets for 1001 patients with chronic lymphatic leukemia (CLL). The mRNA data for this study were from the GEO database [27]. The KEGG networks used to merge for CLL were the Notch signaling pathway, T-cell receptor signaling pathway, and Wnt signaling pathway.

## 3. Discussion

Above, we have described the target selection for the inhibition of protein–protein interactions for patient-specific types of cancer. We have provided two concrete examples: glioma and CLL. Each of these examples results in a diagram showing the most important targets based on the number of patients in the database with similar profiles. In general, the next step in the application of this process is to seek approved or investigational drugs that are able to inhibit these interactions specifically and selectively. This can be accomplished by using either DrugBank [28], ChEMBL [29], KEGG Drugs [30], or other medicinal chemistry databases that are publicly available. In the absence of such inhibitors, there are more demanding computational drug discovery methods that can be brought to bear on the problem, which can lead to the identification of putative inhibitors of the selected protein–protein interactions. These approaches typically require the knowledge of the crystal structure of the interacting proteins, which can be readily inspected using the Protein Data Bank (PDB) [31]. Once these structures are found, following their equilibration using Molecular Dynamics, virtual screening of large medicinal chemistry databases such as ZINC [32] can be performed for putative inhibitors. The results of such searches are useful as initial steps in pre-clinical validation experiments using in vitro assays. Recently, a very accurate machine learning methodology was introduced into the field by DeepMind, which allows us to predict in silico the crystal structure of any amino acid sequence of a protein. The use of AlphaFold [33] has revolutionized the field of computational drug discovery by allowing structure-based drug searches without experimentally known crystal structures of protein targets. Various additional computational methods can be utilized to further refine the search for drug candidates for a specific protein–protein interaction inhibitor. Without attempting an exhaustive review, here we cite some methods, all prior to [12,13,14]. Mashiach et al. [34] describe a web server tool for the flexible induced fit of the molecular backbone in molecular docking. Schymkowitz et al. [35] describe an online molecular force field for studying rapidly the folding dynamics of proteins. Ogman et al. [36] describe a method for predicting protein–protein interactions. These, and many others, could be cited. They all have in common the solution to the problem of matching protein folding with the discovery of a small molecule or peptide that can effectively dock and bind in order to block some protein–protein interaction. Various docking strategies have been discussed in the literature [37], and they focus on the draggability of the targets and drug-likeness of the medicinal chemistry compounds such that the ligand’s specificity against a particular protein target can be calculated to enable further lead optimization processes. Molecular docking programs perform search algorithms whereby the conformation of the ligand is evaluated recursively until the convergence to the minimum energy is reached. In order to quantify the affinity of the ligand for the target, a scoring function is computed, typically a Gibbs binding free energy ΔG, which is then used to rank the candidate poses as the sum of enthalpic contributions, i.e., the electrostatic and van der Waals energies in addition to the entropic part due to removal of water molecules from the protein–drug interface. After a docking campaign is completed with a ranked list of the putative inhibitors, their experimental validation can follow.

## 4. Materials and Method

As briefly discussed in the Introduction, our proposed approach is to again use mRNA from blood or biopsied tissue, normalize the values, and instead of overlaying it on the entire human PPI network, we overlay it on the appropriate KEGG pathway(s) [38]. For example, for glioma cancer, we merged the glioma network, ErbB signaling pathway, and EGR tyrosine kinase inhibitor resistance pathway, as shown in Figure 3.

It should be emphasized here that in spite of some similarities, the current paper uses a distinct methodology compared to the previously published work. The former approach involves essentially removing the protein (vertex) from the network, thus removing all the possible interactions for that protein. This protein is selected by first computing the Gibbs energy for each protein in the entire human PPI network and then performing a topological filtration on the energy landscape. In this smaller network, we then computed the Betti number for the resulting network. We subsequently recomputed the Betti number after removing each protein and selecting the protein that changed the Betti number the most, which reduces the complexity of the network the most.

The new approach starts with the merging of the appropriate KEGG pathways and then computing the Gibbs energy of each protein. The next step in the new algorithm is to find the protein with the largest Gibbs energy. That protein is not to be removed, but rather the interaction with that protein and the neighboring protein with the largest expression value. Consequently, the overall aim is to discover the protein–protein interaction. Accordingly, no Betti number computation is needed in contrast to the previous approach. Below, we describe in detail each step of the new and improved approach.

After the adjacency list of protein–protein interactions from the merged network is obtained (Figure 3), the next step is to find some mRNA relevant to the disease. We downloaded the data of 514 glioma patients from the TCGA database [26]. After normalizing each patient’s mRNA data to be in the range between (0,1), we are ready to calculate the Gibbs energy attributed to each protein (node in the network) in the PPI network according to the formula
(3)$$G_{i}\leftarrow c_{i}ln\frac{c_{i}}{\sum_{j}c_{j}}$$

When the data are rescaled between 0 and 1, it indicates that the most up-regulated protein is set to 1 and the most down-regulated protein is set to 0. The rationale for this is discussed in [8]. In Equation (3), $c_{i}$ represents the concentration of the protein, *i* under consideration. Since we usually do not have knowledge of the exact concentration of the protein, we use the normalized mRNA as a surrogate, which is a good approximation. The denominator is the sum of all the normalized mRNA data for the proteins connected to node *i* and includes the concentration value for *i.* The argument of the *ln* function will be less than 1, so the result is a negative number, representing Gibbs free energy. The more negative the Gibbs free energy value, the more important the contribution of the selected protein *i* to the stability of the PPI network. Technically, Equation (3) is not an equality because the units do not match on each side. We rescale the concentration to protein *I*; thus, the relation is a mapping.

After we have computed the Gibbs energy of each protein, we then select the one that has the largest absolute value, meaning the most negative. This protein contributes the most to the overall energy of the tumor cells. Once that protein is identified, we select the nearest neighbor that has the largest expression or the node that has the individual highest concentration and is also connected to the identified node with the highest absolute Gibbs energy. These two nodes, which are certainly connected to each other, comprise the link to inhibit. In other words, these two proteins are the ones to be targeted for inhibition (to limit their reactivity). Figure 4 shows a small example. Assume we have identified the node RB1 as representing the protein with the largest absolute Gibbs energy. Then, we need to search the nearest neighbors of RB1, which are E2F1, E2F2, E2F3, CCND1, CDK6, and CDK4. Which of those proteins has the largest mRNA expression for this patient? Suppose it is CCND1. That means we want to inhibit the interaction of RB1 with CCND1 but no others. We want to leave the other proteins that RB1 interacts with alone, and we want to leave the other proteins that CCND1 interacts with alone—namely, CDKN1A and CDKN2A.

## 5. Conclusions

In this paper, we have proposed a new method for the optimized, patient-specific selection of drug targets for cancer and possibly other diseases. The method involves the analysis of protein–protein interaction networks and their subsets related to the over-expressed proteins in cancer cells of interest. It then quantifies the importance of the targets for inhibition based on their Gibbs free energy. Subsequent proteins interacting with the target are ranked according to their expression levels. Two specific examples have been evaluated this way: glioma and CLL. We have also discussed how this approach can be further applied in the clinical setting by searching for approved and investigational drugs that inhibit the protein interactions identified in our methodology.

Our results have three major implications for cancer therapy, namely (i) the methodology introduced here defines a quantitatively-based strategy to identify potential oncotargets for any cancer chemotherapy treatment for which PPI networks have been generated; (ii) it identifies and ranks potential targets of inhibition in combination chemotherapy which offer the most promising clinical outcomes; and (iii) it provides rapid identification of molecular targets with existing approved or experimental pharmacological agents in the context of personalized medicine based on individual tumors types and histological subtypes. However, we are well aware of the need for experimental validation of this at both an in vitro and an in vivo level.

With the ever-increasing pace of modern technological development, it is highly probable that in the foreseeable future, a patient diagnosed with cancer will be biopsied, from which a DNA sequence of both malignant and stromal cells will be performed that will subsequently be used to inform the treatment plan. This treatment plan will be based on the generation of a PPI network for both somatic and germline cells, allowing for the computational identification of ranked oncotargets. Furthermore, combining these results with the existence and pharmacological profiles of corresponding oncotarget inhibitors will result in a rationally designed patient-specific combination chemotherapy. This strategy can be further refined by aiming for minimal side effects for normal cells in cases where off-target interactions of the selected inhibitors are known or can be computationally predicted through docking or artificial intelligence approaches. This strategy, when properly validated by in vitro and in vivo assays, is capable of overcoming drug resistance when applied in an iterative, time-dependent manner.

Finally, we hope that this strategy may assist in the development of integrated methods based on a number of fields of science, such as bioinformatics, computational drug discovery, artificial intelligence, and pharmacokinetics, with applications in medicine and healthcare. This trend in the introduction of modern and sophisticated medical-research-related advances will eventually bring tangible clinical benefits to patients, as well as cost-savings and greater efficiencies to healthcare and medical insurance providers. However, it must also be kept in mind that associated clinical, societal, and ethical risks are present when innovation is initially implemented. Therefore, a balanced assessment of the risks and benefits of such strategies must be carefully investigated before clinical introduction [40].

## Figures and Tables

**Figure 1 ijms-24-14648-f001:**
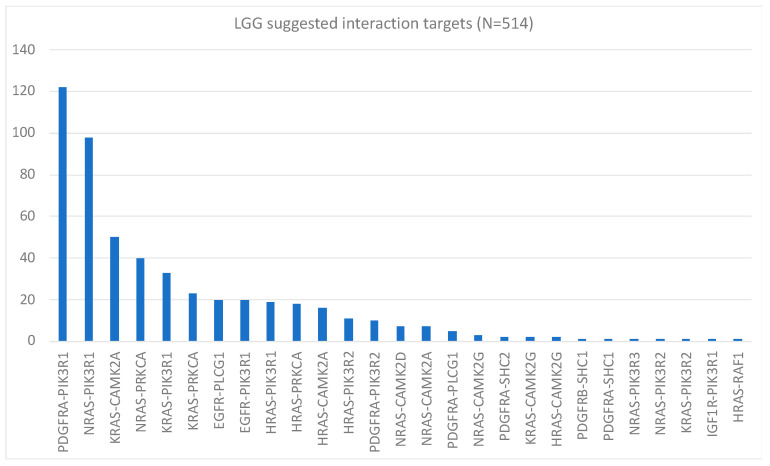
Computed protein–protein interaction targets for glioma cancer patients. The Pareto chart is read as follows: The chart shows that the best inhibition target for just over 120 patients is the PDGFRA-PIK3R1 interaction. It also shows that for 50 patients, the best target is KRAS-CAMK2A. The other bars are interpreted similarly.

**Figure 2 ijms-24-14648-f002:**
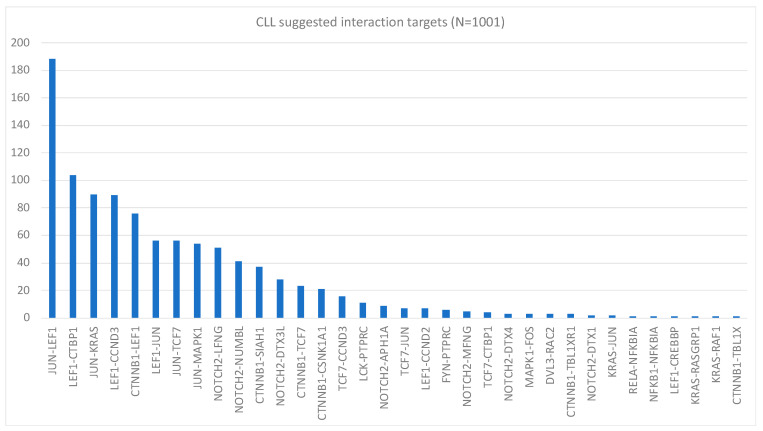
Pareto chart showing the interaction targets for CLL. The chart is interpreted similarly to that shown in Figure 1.

**Figure 3 ijms-24-14648-f003:**
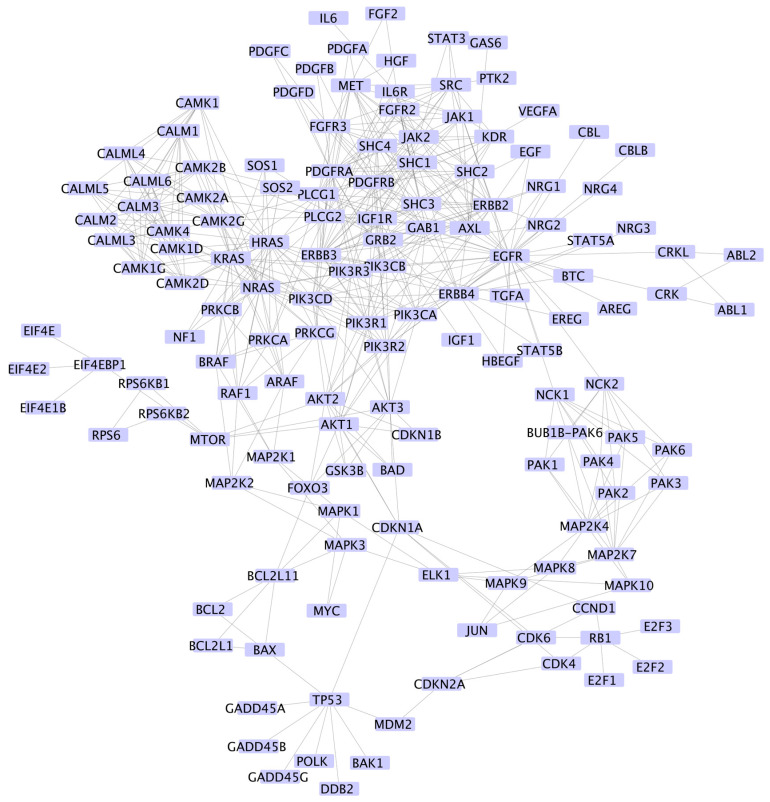
Merged KEGG [38] glioma network, ErbB signaling network, and EGFR tyrosine network merged using Cytoscape [39]. This network contains 144 nodes and 517 edges.

**Figure 4 ijms-24-14648-f004:**
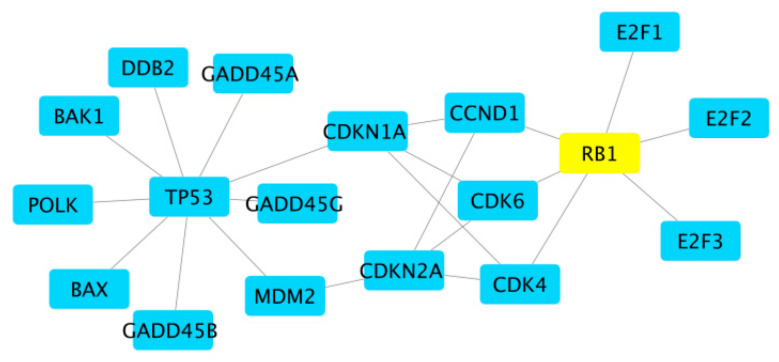
For illustrative purposes, consider this piece of the merged network shown in Figure 3. If, by whatever means, we have identified the RB1 protein to inhibit, that will eliminate its interaction activity with E2F1, E2F2, E2F3, CCND1, CDK6, and CDK4, whereas if we really only wanted to eliminate the activity of RB1 with CCND1, we would need to find a specific molecule that binds to either RB1 or CCND1 in such a way as to block their mutual interaction.

## Data Availability

The data for the glioma patients are from TCGA Website [28]. The data for the CLL study are from the GEO Website and, more specifically, are the following datasets: GSE10139, GSE28654, GSE31048, GSE39671, GSE49896, GSE50006, and GSE69034.

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
