# Peer review of "Gibbs Energy and Gene Expression Combined as a New Technique for Selecting Drug Targets for Inhibiting Specific Protein–Protein Interactions"

_ijms, 2023, doi:10.3390/ijms241914648_

Round 1

Reviewer 1 Report

Rietman, et al. proposed a new method to select the most important protein-protein interaction (PPI) that can be drug target for inhibition. They provided two application examples with potential PPI targets identified by this method and discussed the possible next steps for drug discovery.

The new method is a simplified version of the previously proposed method, where the old method selects the target PPI according to Betti centrality, while the new method directly uses the Gibbs energy and concentration. It is unknown whether this new method can provide reliable results, nor if it has improvements compared with the old one. This work doesn’t provide notable insights in this field. Therefore, I do not recommend publishing this manuscript.

English writting is okay, except for a few typo errors.

Author Response

We thank the reviewer for her/his comments. We have rewritten significant portions of the manuscript in which we have clarified all the issues raised. All the changes are highlighted.

Again thank you.

Reviewer 2 Report

The work concerns a methodology for the optimal selection of a molecular target in anti-cancer therapy. The proposed procedure, based on the combination of protein-protein interaction networks with mRNA expression data, enables the identification of a key protein. Below are my comments:

(1) The work appears to be more like a summary of previous works by the authors on the same topic rather than a full research article. According to the Abstract and Introduction, the current article represents an extension of previously published concepts. Unfortunately, it is not entirely clear what this extension would entail. The methodology summarized in the "Method and Background" section seems to be identical to that proposed earlier in refs. [1] and [2]. If the work contains significant modifications, the authors should clearly and explicitly specify them while comparing them to the previously existing method. Additional examples of other protein-protein interaction networks illustrating the applicability of the method do not provide sufficient justification for publication.

(2) The authors use the term "Gibbs free energy" in a different context than the thermodynamic definition of this state function, which should be clearly indicated.

(3) Minor things: there are several typos throughout the manuscript, including Abstract (e.g. “Gibb free energy”, “cases ,glioma and CLL,”. Moreover, I suggest to explain all abbreviations at their first occurrences.

Except of some typos, the quality of English is very good.

Author Response

(The authors gave the same response as above.)

Round 2

Reviewer 1 Report

This revised manuscript has resolved all my concerns. Therefore I recommend publishing this version.

English writting is good.

Author Response

Thank you for your time.

Reviewer 2 Report

The revised version of the paper clearly highlights the differences between the original approach published earlier and the new method described now. The revisions also emphasize the novelty aspect in the current work. It appears that the value of the results could be further enhanced by including a comparison not only of the methodologies but also of the specific outcomes for a selected system obtained using both versions of the method. Apart from this, I have no further comments.

Author Response

Thank you for your time and suggestions on improving the manuscript. We agree with you that a direct comparison between the two approaches would be very interesting; however, we what you are proposing would involve, at the least, an in vitro batch of experiments. That would  go beyond the scope of this paper.

Again thank you for the review.